# Epidemiological Profile and Social Welfare Index as Factors Associated with COVID-19 Hospitalization and Severity in Mexico City: A Retrospective Analysis

**DOI:** 10.3390/ijerph192214803

**Published:** 2022-11-10

**Authors:** Mario Antonio Téllez-González, Juan Antonio Pineda-Juárez, Juan Antonio Suárez-Cuenca, Mónica Escamilla-Tilch, Daniel Santillán-Cortez, Silvia García, Sofía Lizeth Alcaraz-Estrada, Juan Carlos Pérez-Razo, Carlos Alberto Delgado-Quintana, Joel Vargas-Hernández, Sandra Muñoz-López, Maricela Escarela-Serrano, Maribel Santosbeña-Lagunes, Alejandro Alanís-Vega, Ricardo Platón Vázquez-Alvarado, José Alfredo Merino-Rajme, Paul Mondragón-Terán

**Affiliations:** 1Centro Médico Nacional “20 de Noviembre”, Coordination of Research, Mexico City 03229, Mexico; 2Centro Médico Nacional “20 de Noviembre”, Clinical Research Service, Mexico City 03229, Mexico; 3Centro Médico Nacional “20 de Noviembre”, Genomics Research Division, Mexico City 03229, Mexico; 4Centro Médico Nacional “20 de Noviembre”, COVID-19 Group, Mexico City 03229, Mexico

**Keywords:** COVID-19, social welfare, healthcare disparities, socioeconomic factors, Mexico

## Abstract

Epidemiological data indicate that Mexico holds the 19th place in cumulative cases (5506.53 per 100,000 inhabitants) of COVID-19 and the 5th place in cumulative deaths (256.14 per 100,000 inhabitants) globally and holds the 4th and 3rd place in cumulative cases and deaths in the Americas region, respectively, with Mexico City being the most affected area. Several modifiable and non-modifiable risk factors have been linked to a poor clinical outcome in COVID-19 infection; however, whether socioeconomic and welfare factors are associated with clinical outcome has been scanty addressed. This study tried to investigate the association of Social Welfare Index (SWI) with hospitalization and severity due to COVID-19. A retrospective analysis was conducted at the Centro Médico Nacional “20 de Noviembre”—ISSSTE, based in Mexico City, Mexico. A total of 3963 patients with confirmed or suspected COVID-19, registered from March to July 2020, were included, retrieved information from the Virology Analysis and Reference Unit Database. Demographic, symptoms and clinical data were analyzed, as well as the SWI, a multidimensional parameter based on living and household conditions. An adjusted binary logistic regression model was performed in order to compare the outcomes of hospitalization, mechanical ventilation requirement (MVR) and mortality between SWI categories: Very high (VHi), high (Hi), medium (M) and low (L). The main findings show that lower SWI were independently associated with higher probability for hospital entry: VHi vs. Hi vs. M vs. L-SWI (0 vs. +0.24 [OR = 1.24, CI_95%_ 1.01–1.53] vs. +0.90 [OR = 1.90, CI_95%_ 1.56–2.32] vs. 0.73 [OR = 1.73, CI_95%_ 1.36–2.19], respectively); Mechanical Ventilation Requirement: VHi vs. M vs. L-SWI (0 vs. +0.45 [OR = 1.45, CI_95%_ 1.11–1.87] vs. +0.35 [OR = 1.35, CI_95%_ 1.00–1.82]) and mortality: VHi vs. Hi vs. M (0 vs. +0.54 [OR = 1.54, CI_95%_ 1.22–1.94] vs. +0.41 [OR = 1.41, CI_95%_ 1.13–1.76]). We concluded that SWI was independently associated with the poor clinical outcomes in COVID-19, beyond demographic, epidemiological and clinical characteristics.

## 1. Introduction

The speed of spread and severity of COVID-19 (SARS-CoV-2 infection) worldwide was alarming due to the clinical (mostly respiratory) symptoms and poor prognosis outcomes, such as hospitalization or mortality [1,2,3], as well as becoming a public health emergency of international concern, reaching the category of: pandemic [4,5]. To date (October 2022), confirmed cumulative cases have reached approximately 620 million and 6.5 millions of deaths worldwide [4].

According to the World Health Organization (WHO), to October 2022, Mexico holds the 19th place in cumulative cases (5506.53 per 100,000 inhabitants) and the 5th place in cumulative deaths (256.14 per 100,000 inhabitants) globally and as well as the 4th and the 3rd places in cumulative cases and deaths (respectively) in the Americas region. Globally, there have been over 7 millions of cases of contagion and 330,000 deaths [6,7].

The burden of COVID-19 in Mexico has largely impacted in diverse aspects. Factors such as comorbidities (diabetes, hypertension, obesity), non-healthy behaviors (smoke) and non-modifiable factors (sex or age) have been largely associated to a higher number of cases with COVID-19 and higher hospitalization rate in Mexican population, similar to other populations [8,9,10,11,12].

Social factors also are associated with the risk of SARS-CoV2 infection and COVID-19 clinical prognosis. Some studies have suggested that social disparities and determinants of health could explain why people living in areas with high poverty levels and with informal/basic jobs are less likely to maintain social distancing. Interestingly, ethnicity (Black and Hispanic patients) both associated to higher hospitalization rates but without differences in mortality rates due to COVID-19 [13,14]. In addition, people living in conditions of socioeconomic deprivation, such as crowded populations, are associated with lower healthcare service access, low income, increased number of persons per house, and/or unhealthy lifestyles factors, leading to higher rates of SARS-CoV2 infection and possibly worse COVID-19 outcomes, such as the need of hospitalization or the need for mechanical ventilation, understood as a condition of intensive care unit hospitalization. [15,16]. Other socioeconomic factors, such as low-income jobs or unemployment (i.e., deriving in lower opportunities to purchase protective equipment, such as appropriate facemasks), low educational level (deriving in not understanding or not following health recommendations or the correct use of the protective equipment), poor diet quality or the use of highly crowded public transport are linked to COVID-19. [17,18,19,20].

It is widely known that countries with higher economic income and little social inequalities are more probable to maintain a better control and stability of pandemic impact [21]. However, within Mexico, Mexico City shows a complex situation regarding social care. A significant proportion of the Gross Domestic Product (GDP) is assigned to Mexico City’s population and their socioeconomical requirements [22]; meanwhile, there are deep differences in the socioeconomic and social services access among the 16 major regions constituting Mexico City. Such differences may be objectively evaluated by measures like the Social Welfare Index (SWI), a score based in the economic, social and environmental factors [23].

Although socioeconomic factors potentially impact to SARS-CoV2 infection and COVID-19 clinical behavior, particularly in a city with the crowded characteristics of Mexico City [24], welfare factors association with COVID-19 impact in clinical outcome has scanty been addressed.

## 2. Materials and Methods

### 2.1. Design

A retrospective study was conducted at the Centro Médico Nacional (CMN) “20 de Noviembre” of the Instituto de Seguridad y Servicios Sociales de los Trabajadores del Estado (I.S.S.S.T.E.), a tertiary level hospital based in Mexico City. The collected variables for this study were classified as follows; main outcome variables: hospitalization, mechanical ventilation requirement and mortality, while the main independent variable was the SWI; meanwhile, sex, age, number of comorbidities and smoke history were defined as covariates/confounders. In addition, hospital admission service, clinical and type of treatment data were collected.

### 2.2. Data Collection

Demographic (sex, age and major of precedence) and clinical-epidemiological data, such as symptoms, comorbidities and primary clinical outcomes (hospitalization, mechanical ventilation requirement and mortality) were retrieved from the Virology Analysis and Reference Unit database (VARU) based at the CMN “20 de Noviembre”. This database contains information from patients nested within hospital from admission to discharge or negative outcome (mortality) with confirmed or suspected COVID-19 diagnosis and whose address was located in Mexico City and were registered in the database from March to July 2020. Confirmed COVID-19 infection were tested by Real Time-Polymerase Chain Reaction and suspected COVID-19 infection were defined by clinical symptoms as person with cough, fever or headache accompanied at least one of the following signs or symptoms: dyspnea, arthralgia, myalgia, odynophagia/pharyngeal burning, rhinorrhea, conjunctivitis or chest pain in the last 7 days, according to the Epidemiological Vigilance National Committee [25].

The SWI was taken based on location for each of the 16 major areas that integrate Mexico City: Álvaro Obregón, Azcapotzalco, Benito Juárez, Coyoacán, Cuajimalpa, Cuauhtémoc, Gustavo A. Madero, Iztacalco, Iztapalapa, Magdalena Contreras, Miguel Hidalgo, Milpa Alta, Tláhuac, Tlalpan, Venustiano Carranza and Xochimilco) [18]. The 11 elements that integrate the SWI were obtained from the annual report from the Council for the Evaluation of Social Development of Mexico City [18], which included objective indicators (education, employment, health, income, food and household conditions), subjective (satisfaction with life and happiness), and others, such as social cohesion, use of technology, access to culture and recreation and quality of the physical environment. Then, major areas were stratified into 4 categories according to their SWI (Figure 1), as follows: Very High-SWI (VH-SWI): Benito Juárez, Miguel Hidalgo, Azcapotzalco and Coyoacán; High-SWI (H-SWI): Cuauhtémoc, Venustiano Carranza, Iztacalco and Álvaro Obregón; Medium-SWI (M-SWI): Magdalena Contreras, Gustavo A. Madero, Cuajimalpa and Iztapalapa; and Low-SWI (L-SWI): Tlalpan, Tláhuac, Xochimilco and Milpa Alta. Selection of cases for information analyses is shown in the flow chart of analysis (Figure 2). Figure 3 shows the exponential growth of total positive and cumulative cases and per 100,000 inhabitants, and Figure 4 shows the exponential growth of COVID-19 cases per major divided by SWI categories and per 100,000 inhabitants.

### 2.3. Data Analytic Strategy

Continuous data are presented as median and 25–75 percentiles and categorical data as frequencies and percentages. In order to evaluate the differences between the SWI categories, the Kruskal–Wallis test was used for continuous data and Pearson’s chi-squared test for categorical data. A binary logistic regression model was performed in order to determinate the association between the SWI categories and COVID-19 clinical outcomes and also to adjust the models considering potential confounder factors (sex, age, number of comorbidities and smoking history). VH-SWI was taken as category reference in all models. Also, to estimate the good model fit of logistic regression, the Omnibus, −2 Log Likelihood, AIC and Nagelkerke’s R^2^ tests were evaluated. *p* values < 0.05 were considered statistically significant. All analyses were performed using IBM SPSS Statistics v24 (IBM SPSS Statistics for Macintosh, Version 24.0. Armonk, NY, USA: IBM Corp.).

Sample size was not calculated for any primary clinical outcomes, however statistical power was calculated for each final adjusted logistic regression model (model 2) using the software G*Power^®®^ [26] in accordance to a two tailed post hoc test with the following parameters: hospitalization outcome; total sample size = 3963, precision = 0.05, R^2^ = 0.153, odds ratios = 1.24, 1.90, 1.73, mechanical ventilation requirement outcome; total sample size = 3963, precision = 0.05, R^2^ = 0.023, odds ratios = 1.24, 1.45, 1.35; mortality outcome; total sample size = 3963, precision = 0.05, R^2^ =0.094, odds ratios = 1.54, 1.41, 1.02; all the calculated statistic power were >99%. The database does not contain missing values.

## 3. Results

The study population was constituted by 3963 cases, median age was 45 years old, mainly distributed within the range of 19–59 years old (73%), whereas male gender were predominant (55.3%). Similar characteristics were observed between SWI categories. Previous exposure with a COVID-19 positive case was reported in 40.6%, with higher frequency in the LSWI category (50.2%, *p* < 0.01). Related to the results of the diagnosis of SARS-CoV-2, 52.2% were classified with positive test and 44.1% with clinical-tomographic evidence to SARS-CoV2 infection. The rest of the demographic and clinical variables, as well as clinical outcomes are presented in Table 1 and Table 2.

Most prevalent comorbidities were systemic arterial hypertension (29.9%), diabetes mellitus (25.9) and obesity (22.8%), followed by other comorbidities of lower prevalence, such as chronic obstructive pulmonary disease (COPD, 4.3%), chronic renal disease (5.3%), cardiovascular disease (4.4%), autoimmune diseases (3.3%) and asthma (2.8%). Likewise, most of the patients reported no comorbidities (41.2), one comorbidity (27.6%), two comorbidities (17.9%), or three (13%).

When analyzed by SWI categories, H-SWI and M-SWI showed higher prevalence of diabetes mellitus and hypertension (diabetes mellitus 26.2% and 26.9%, *p* < 0.03; hypertension 32.7% and 30%, *p* < 0.01, respectively), as compared to lower SWIs, being this last category was the most prevalent for obesity (29.6%, *p* < 0.001). Likewise, the number of comorbidities showed heterogeneous distribution between SWIs. The H-SWI category reported three or more comorbidities as considerably prevalent (15.8%), while SWI the L-SWI category reported a lack of comorbidities as the most prevalent (37.4%; *p* = 0.03). In addition, smoking history was more prevalent in the L-SWI category (17.2%, *p* < 0.001) (Table 1).

After initial medical assessment, the risk to be hospitalized due to COVID-19 increased as the SWI reduced (29% for H-SWI vs. 84% and 71% for M-SWI and L-SWI respectively, Table 3). Regarding the indication for hospitalization, most cases were attended due to severe acute respiratory distress (52.3%), followed by non-severe cases attended at triage (18.7%) and cases already admitted for a medical condition different from SARS-CoV2 but with resulting positive in a protocol test for COVID-19 (26.8%).

Most prevalent presenting symptoms were cough (75.1%), fever (71.8), headache (69.7%), acute respiratory distress syndrome (ARDS, 65.1%) and pneumonia (60.4%); followed by dyspnea, myalgia, arthralgia and others. Significative differences in the prevalence of most symptoms was found between SWI categories. Besides standard COVID-19 therapy based on anti-pyretic drugs and steroid, antiviral or antibiotic agents were administered in 21.1% and 60.2% of the cases, respectively (Table 2).

During hospitalization, primary clinical outcome of severity was characterized by requirement of mechanical ventilation (16%) and mortality (22.1%). Mechanical ventilation was more prevalent between M-SWI and LSWI (24.2% and 22.8%, respectively, *p* < 0.05), whereas mortality was more frequent between M-SWI and H-SWI and (23.7% and 25.8%, *p* < 0.01) (Table 2). Such clinical outcomes were affected by the type and number of comorbidities.

Furthermore, clinical severity was independently associated to L-SWI, as much as two-fold risk (OR = 2.07), either in non-adjusted or adjusted models; while mortality rate was associated H-SWI (59%) and M-SWI (42%) but modified after adjustments by age, sex and co-morbidities (model 1: 57% and 41%; model 2: 54%, 41%, respectively; Table 3).

## 4. Discussion

As far as we know, this study is the only one study that has evaluated the association of COVID-19 outcomes with the social impact measured by an index based in the economic, social and environmental factors (SWI) considering a high populated city, such as Mexico City. Interestingly, higher SWI were related with higher number of comorbidities, and lower SWI was only associated with obesity. Furthermore, the probability to be hospitalized and require mechanical ventilation increased as the SWI reduced. Conversely, mortality was more frequent between medium and high SWI, thus reflecting the heterogeneous distribution of population between SWIs, probably involving pathophysiological and/or immunological profiles.

Epidemiological profiles of COVID-19 in Mexican population, as reported in different series, have been very consistent regarding demographic behavior characterized by a higher proportion in males and most of cases middle aged [8,27,28]. Co-morbidities such as hypertension and diabetes mellitus were frequently observed in our series, with some difference in proportions as compared to other studies [8,27,28]. Interestingly, we observed a 22.8% prevalence of obesity, which is lower than that reported by other series [27,28] and may be due to selection bias, since some communities may be underrepresented, while a potential effect related to a lower SWI, as suggested by the comparative analysis (Table 1) could not be ruled out.

Of note, our data analyses involved information available from the General Department of Epidemiology of the Mexican Ministry of Health, and our results are comparable with data reported from other tertiary health centers in Mexico City, for example, Instituto Nacional de Ciencias Médicas y Nutrición “Salvador Zubirán” (INCMNSZ) [29]. Series from this last health institution included 309 patients with diagnosis of COVID-19, similar in age and sex to our study population but different in prevalence of hypertension, obesity, diabetes mellitus, COPD and CVD (CMN “20 de Noviembre” vs. INCMNSZ: 20.9% vs. 19.7%; 22.8% vs. 39.6%; 25.5% vs. 13.3%; 4.3% vs. 1% and 4.3% vs. 2.9%, respectively), probably due to differences in the administrative, social and worker-related requirements for medical attention at each given institution. Prevalence of presenting symptoms like fever, cough, chest pain, dyspnea, myalgia, arthralgia and headache were similar. This pandemic behavior requires such a detailed description as it was performed in 2009 during pandemic due to influenza virus H1N1 [30].

Most prevalent comorbidities included systemic arterial hypertension, diabetes mellitus and obesity, which is consistent with previous observational studies [8,9,31,32], which also suggest their role as risk factors associated to poor clinical outcomes in COVID-19 patients [31,32,33].

As we mentioned, the plausibility that socioeconomic factors contribute to the poorest outcomes for COVID-19 beyond of the epidemiological-clinical characteristics suppose something to consider. Results of COVID-19 observed in this study can be compared with some studies conducted in different population with similar results when social indexes and variables of socioeconomic status (SES) were considered. For example, Foster et al. [15] reported major risk of severe cases (RR 6.02, 95% CI 4.72–7.71) and higher mortality (RR 9.60, 95% CI 4.70–21.44) in individuals living in deprivation areas and had unhealthy life styles in United Kingdom. Likewise, specific data of SES as poverty level, income, education level, household size and ethnicity has been associated with higher probability of mortality and hospitalization rate in more than 10% to almost 80% for USA reports [14,16]

Furthermore, our data were similar to others few studies conducted in Mexico. The results reported by Bello et al., [34] in Mexican population using 2015 Social Lag Index (SLI, multicomponent formed by socioeconomic factors, home living and access to basic services) addressed to estimates social disadvantage and structural inequality, finding an association with higher SLI and COVID-19 severity and lethality cases but only in elder population (HR 1.13, 95% CI 1.05–1.21, adjusted by age and sex). In addition, the authors found an association with SLI and some COVID-19 outcomes related to respiratory distress (OR 1.45, 95% CI 1.21–1.74) hospitalization entry (OR 1.52, 95% CI 1.24–1.87) and mechanical ventilation requirement (OR 1.45, 95% CI 1.21–1.87). Similarly, Antonio et al., [35] used the 2020 SLI independent of density population (DISLI) to assess the association with COVID-19 outcomes in Mexico City population. The results proved that the higher DISLI, the higher the risk of mortality (IRR 2.42, 95% CI 1.03–5.72) due to suspected cases of COVID-19. In addition, a major risk of hospitalization, severe cases and mortality were found when the DISLI and high population density were higher. To date, this last study is the unique of its kind with similar objective and location (Mexico City) to ours. However, some differences can be stressed in order to improve the actual knowledge; for example, SLI is a National and multidimensional index developed by the Mexican National Council for the Evaluation of Social Development Policy, which includes indictors of education level, access to health services, basic services and household characteristics to assess social deprivation; meanwhile, the SWI (a specific index at local level) adds more components as satisfaction with life and happiness, social cohesion, use of technology, access to culture and recreation and quality of the physical environment. Thus, incorporating this information to current knowledge could close the gap between the “socioeconomic status and COVID-19 outcomes”, specifically in individuals living in Mexico City as an example of overcrowded city.

Finally, our results were similar to data reported by Ortíz et al., [36] where the cases with COVID-19 from Mexican South Region (with a higher poverty index and large proportions of indigenous ethnic groups) showed a higher risk for hospitalization and mechanical ventilation requirement than a population from the Center or North regions of the country. Similarly, cases with COVID-19 in urban areas lacking public healthcare affiliation were associated with worse outcomes [37] as well as municipal poverty. [38]. In addition, socioeconomic inequalities also give rise to unfollowing of preventive behaviors (such as home office strategy, hand washing, recurrently use of face mask and sanitizers) against COVID-19, as reported by Irigoyen et al., wherein people with a medium income were more likely (two-fold-increased probability) to carry out the health recommendations in comparison with people with a lower income, [39] hence increasing the risk of contagions.

Some limitations of the present study include the study design and the heterogeneous data source. Meanwhile, the use of a limited sample population was considering the inclusion of clinical COVID-19 outcomes, data of comorbidities and SARS-CoV2 PCR molecular diagnosis. Another limitation is that in order to obtain specific differences by region, a deeper analysis and data of the 16 majors regions in Mexico City would be necessary. Nevertheless, this study provides results that closely reflect the real scenario of a population with heterogeneous socioeconomic characteristics and their epidemiological impact to address the pandemic of COVID-19.

## 5. Conclusions

Our study demonstrates the association between SWI and clinical outcomes related to COVID-19. Interestingly, we found that lower SWI categories were associated with a higher the risk of mortality, hospitalization and mechanical ventilation requirement beyond demographic, epidemiological and clinical characteristics in the Mexico City population; suggesting the relevance to include socioeconomic and healthcare inequalities as additional variables with potential impact in clinical outcomes due to SARS-CoV2 infection. Furthermore, the utility of this information can encourage conducting further studies aimed at decreasing both social and economic differences but also to improve the social disparities in emerging healthcare situations similar to COVID-19 pandemic.

## Figures and Tables

**Figure 1 ijerph-19-14803-f001:**
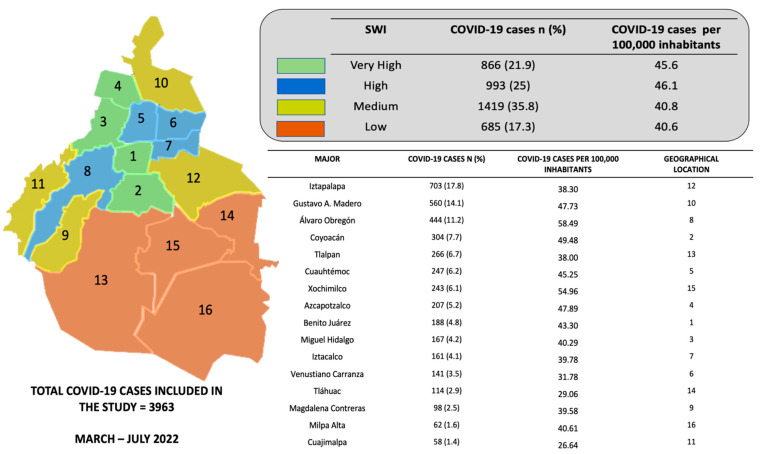
Total COVID-19 cases (n, %) and COVID-19 cases per 100,00 inhabitants in the 16 majors in Mexico City and by Social Welfare Index.

**Figure 2 ijerph-19-14803-f002:**
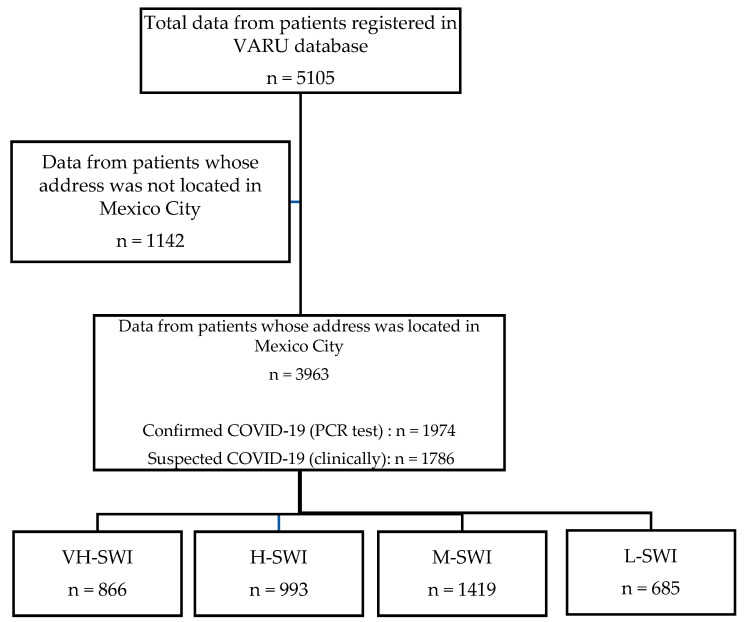
Flow chart of the data entering in the analysis.

**Figure 3 ijerph-19-14803-f003:**
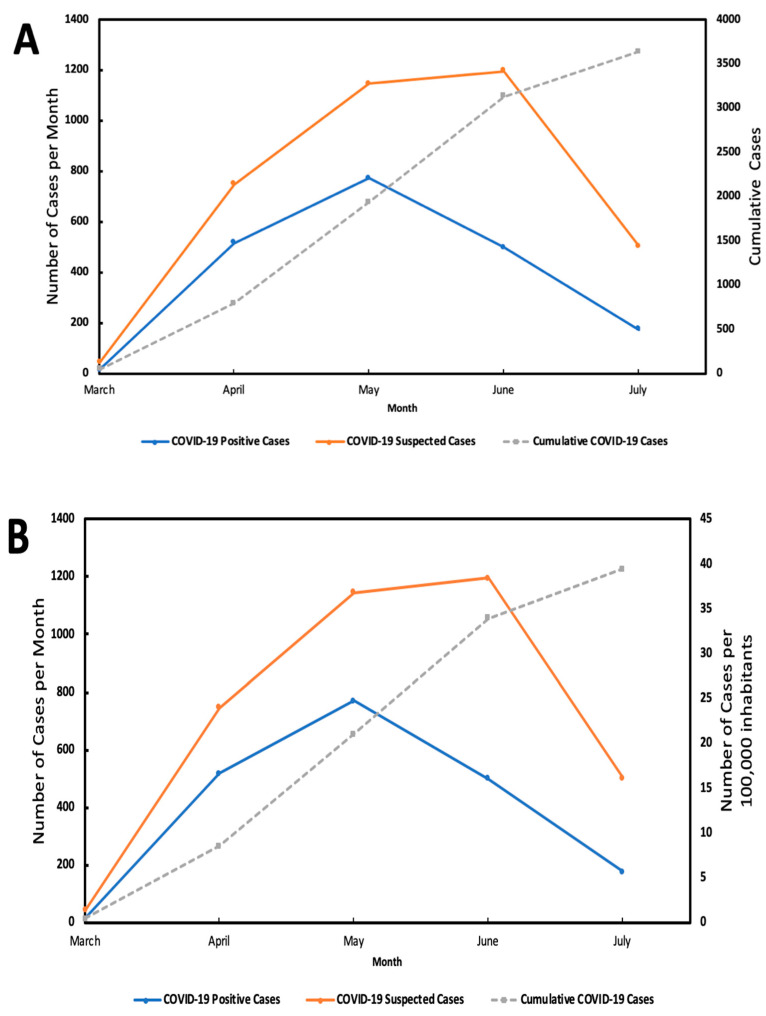
Exponential growth and cumulative cases related to positive (RT-PCR test) and suspected COVID-19 cases: (**A**) represents total cases included from Mexico City and (**B**) represents the number of cases per 100,000 inhabitants, between March–July 2020.

**Figure 4 ijerph-19-14803-f004:**
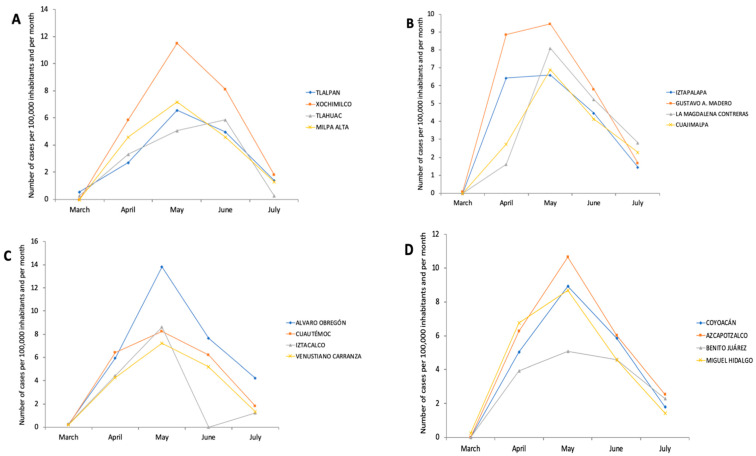
Exponential growth of COVID-19 cases per major divided by SWI categories and per 100,000 inhabitants, between March–July 2020. (**A**)—Low-SWI, (**B**)—Medium-SWI, (**C**)—High-SWI, (**D**)—Very High-SWI.

**Table 1 ijerph-19-14803-t001:** Epidemiological profile and hospital admission service of total sample and stratified by Social Welfare Index categories of patients with COVID-19.

Characteristics	All n = 3963	VH-SWI n = 866	H-SWI n = 993	M-SWI n = 1419	L-SWI n = 685	*p*
Demographic						
Sex n (%)						
Female	1770 (44.7)	396 (45.7)	429 (43.2)	636 (44.8)	309 (45.1)	0.72
Male	2193 (55.3)	470 (54.3)	564 (56.8)	783 (55.2)	376 (54.9)	
Age (years)	45 (29–56)	43 (29–55)	45 (29–56)	46 (29–57)	45 (30–57)	0.04
Age n (%)						
<19 years	339 (8.6)	84 (9.7)	83 (8.4)	125 (8.8)	47 (6.9)	0.12
19–59 years	2900 (73.2)	647 (74.8)	732 (73.7)	1015 (71.6)	506 (74.0)	
>60 years	721 (18.2)	134 (15.5)	178 (17.9)	278 (19.6)	131 (19.2)	
Comorbidities n (%)						
Diabetes	1011 (25.5)	204 (23.6)	260 (26.2)	382 (26.9)	165 (24.1)	0.03
Hypertension	1186 (29.9)	244 (28.2)	325 (32.7)	425 (30.0)	192 (28.0)	<0.01
Obesity	903 (22.8)	176 (20.3)	215 (21.8)	309 (21.8)	203 (29.6)	<0.001
COPD	169 (4.3)	38 (4.4)	54 (5.4)	45 (3.2)	32 (4.7)	<0.01
Asthma	110 (2.8)	27 (3.1)	31 (3.1)	29 (2.0)	23 (3.4)	0.01
Immunosuppression	130 (3.3)	21 (2.4)	34 (3.4)	48 (3.4)	27 (3.9)	0.21
HIV/AIDS	20 (0.5)	7 (0.8)	8 (0.8)	2 (0.1)	3 (0.4)	0.01
Cardiovascular Disease	174 (4.4)	38 (4.4)	42 (4.2)	51 (3.6)	43 (6.3)	<0.01
Chronic Kidney Disease	210 (5.3)	44 (5.1)	48 (4.8)	75 (5.3)	43 (6.3)	0.11
Others	146 (3.7)	27 (3.1)	42 (4.2)	46 (3.2)	31 (4.5)	0.05
No. of comorbidities n (%)						
0	1634 (41.2)	369 (42.7)	411 (41.4)	598 (42.1)	256 (37.4)	
1	1092 (27.6)	231 (26.7)	266 (26.8)	378 (26.6)	217 (31.7)	0.03
2	710 (17.9)	158 (18.2)	159 (16.0)	262 (18.5)	131 (19.1)	
>3	527 (13.3)	108 (12.5)	157 (15.8)	181 (12.8)	81 (11.8)	
Pregnancy n (%)	19 (0.5)	4 (1.0)	2 (0.5)	8 (1.3)	5 (1.6)	0.68
Smoke history n (%)	514 (13.0)	125 (14.4)	118 (11.9)	153 (10.8)	118 (17.2)	<0.001
HAS n (%)						
Triage	741 (18.7)	199 (23.0)	184 (18.5)	228 (16.1)	130 (19.0)	
Medical specialty	1064 (26.8)	225 (26.0)	249 (25.1)	398 (28.0)	192 (28.0)	
Emergency	2073 (52.3)	431 (49.8)	542 (54.6)	752 (53.0)	348 (50.8)	
Pediatric ICU’s	35 (0.9)	3 (0.3)	3 (0.3)	18 (1.3)	11 (1.6)	<0.001
Adult ICU’s	50 (1.3)	8 (0.9)	15 (1.5)	23 (1.6)	4 (0.6)	

Note: Continuous data are presented as median (p25–p75) and categorical data as n (%). VH-SWI—Very High Social Welfare Index, H-SWI—High Social Welfare Index, M-SWI—Medium Social Welfare Index, L-SWI—Low Social Welfare Index. HIV/AIDS—human immunodeficiency virus/acquired Immune Deficiency Syndrome, HAS—hospital admission service, ICU—intensive care unit.

**Table 2 ijerph-19-14803-t002:** Epidemiological-clinical profile, treatment and outcomes of total sample and by Social Welfare Index categories of patients with COVID-19 clinical manifestations.

Characteristics	All n = 3963	VH-SWI n = 866	H-SWI n = 993	M-SWI n = 1419	L-SWI n = 685	*p*
Symptoms n (%)						
Pneumonia	2392 (60.4)	450 (52.3)	631 (63.7)	938 (66.3)	373 (54.6)	<0.001
ARDS	2578 (65.1)	544 (62.8)	658 (66.3)	1014 (71.5)	362 (52.8)	<0.001
Fever	2846 (71.8)	586 (67.7)	685 (69.0)	1117 (78.7)	458 (66.9)	<0.001
Cough	2977 (75.1)	610 (70.4)	738 (74.3)	1152 (81.2)	477 (69.6)	<0.001
Dyspnea	2342 (59.1)	445 (51.4)	599 (60.3)	927 (65.3)	371 (54.2)	<0.001
Odynophagia	1411 (35.6)	270 (31.2)	364 (36.7)	573 (40.4)	204 (28.8)	<0.001
Irritability	743 (18.7)	139 (16.1)	161 (16.2)	285 (20.1)	158 (23.1)	<0.01
Diarrhea	908 (22.9)	178 (20.6)	229 (23.1)	334 (23.5)	167 (24.4)	0.20
Chest pain	1296 (32.7)	255 (29.4)	312 (31.4)	498 (35.1)	231 (33.7)	<0.001
Tremble	1368 (34.5)	236 (27.3)	332 (33.4)	544 (38.3)	256 (37.4)	<0.001
Headache	2763 (69.7)	595 (68.7)	675 (68.0)	1045 (73.6)	448 (65.4)	<0.001
Myalgia	2055 (51.9)	415 (47.9)	529 (53.3)	782 (55.1)	329 (48.0)	<0.001
Arthralgia	2019 (50.9)	426 (49.2)	505 (50.9)	774 (54.5)	314 (45.8)	<0.001
Rhinorrhea	1038 (26.2)	209 (24.1)	257 (25.9)	406 (28.6)	166 (24.2)	<0.001
Polypnea	687 (17.3)	106 (12.2)	167 (16.8)	317 (22.3)	97 (14.2)	<0.001
Vomiting	379 (9.6)	72 (8.3)	99 (10.0)	145 (10.2)	63 (9.2)	<0.001
Abdominal pain	607 (15.3)	130 (15.0)	138 (13.9)	223 (15.7)	116 (16.9)	<0.001
Conjunctivitis	549 (13.9)	96 (11.1)	125 (12.6)	213 (15.0)	115 (16.8)	<0.001
Cyanosis	433 (10.9)	85 (9.8)	94 (9.5)	192 (13.5)	62 (9.1)	<0.001
Anosmia	203 (5.1)	31 (3.6)	57 (5.7)	67 (4.7)	48 (7.0)	<0.001
Dysgeusia	203 (5.1)	34 (3.9)	52 (5.2)	68 (4.8)	49 (7.2)	<0.01
General health discomfort	1934 (48.8)	402 (46.4)	486 (48.9)	757 (53.3)	289 (42.4)	<0.001
SOS	1848 (46.7)	432 (49.9)	471 (47.4)	741 (52.2)	205 (29.9)	<0.001
AV treatment n (%)	837 (21.1)	151 (18.3)	154 (16.0)	462 (33.7)	70 (10.3)	<0.001
AB treatment n (%)	2364 (60.2)	520 (60.8)	593 (60.5)	867 (61.6)	384 (56.4)	0.14
CPC n (%)	1609 (40.6)	357 (47.9)	399 (43.3)	548 (42.4)	305 (50.2)	<0.01
RT-PCR test n (%)						
Positive	1974 (52.5)	402 (48.5)	521 (55.7)	731 (54.2)	320 (49.4)	
Suspected	1657 (44.1)	410 (49.5)	376 (40.2)	570 (42.3)	301 (46.5)	<0.01
NA	129 (3.4)	17 (2.1)	38 (4.1)	47 (3.5)	27 (4.2)	
Clinical Outcomes n (%)						
Ambulatory	1080 (27.3)	303 (35.0)	292 (29.4)	321 (22.6)	164 (23.9)	<0.001
Hospitalization	2883 (72.7)	563 (65.0)	701 (70.6)	1098 (77.4)	521 (76.1)	
Mechanical Ventilation	635 (16.0)	100 (17.8)	151 (21.6)	265 (24.2)	119 (22.8)	0.02
Severity Cases	712 (18.0)	132 (61.4)	176 (63.3)	295 (67.5)	109 (43.4)	<0.001
Mortality	876 (22.1)	155 (17.9)	256 (25.8)	337 (23.7)	128 (18.7)	<0.001
Recovery	832 (21.0)	191 (39.1)	178 (39.1)	323 (50.5)	140 (46.2)	<0.001
HMO	1054 (26.6)	297 (60.9)	277 (60.9)	317 (49.5)	163 (53.8)	

Note: Continuous data are presented as median (p25–p75) and categorical data as n (%). VH-SWI—Very High Social Welfare Index, H-SWI—High Social Welfare Index, M-SWI—Medium Social Welfare Index, L-SWI—Low Social Welfare Index. SOS—sudden onset of symptoms, AV—antiviral, AB—antibacterial, CPC—case person contact, RT-PCR—real-time polymerase chain reaction, HMO—home monitoring observation.

**Table 3 ijerph-19-14803-t003:** Association of the Social Welfare Index with COVID-19 clinical outcomes.

Hospitalization						
Variable	Non-Adjusted Model		Model 1		Model 2	
	OR	95%CI	OR	95%CI	OR	95%CI
VH-SWI	1.00	---	1.00	---		---
H-SWI	1.29	1.06–1.57	1.27	1.04–1.55	1.24	1.01–1.53
M-SWI	1.84	1.52–2.21	1.85	1.53–2.23	1.90	1.56–2.32
L-SWI	1.71	1.36–2.14	1.73	1.37–2.15	1.73	1.36–2.19
Mechanical Ventilation						
Variable	Non-Adjusted Model		Model 1		Model 2	
VH-SWI	1.00	---	1.00	---	1.00	---
H-SWI	1.26	0.95–1.68	1.24	0.93–1.65	1.24	0.93–1.64
M-SWI	1.47	1.13–1.90	1.43	1.12–1.86	1.45	1.11–1.87
L-SWI	1.36	1.01–1.83	1.34	0.99–1.81	1.35	1.00–1.82
Severity Cases						
Variable	Non-Adjusted Model		Model 1		Model 2	
VH-SWI	1.00	---	1.00	---	1.00	---
H-SWI	0.92	0.63–1.33	0.90	0.62–1.30	0.90	0.62–1.31
M-SWI	0.76	0.54–1-07	0.75	0.53–1.06	0.75	0.53–1.06
L-SWI	2.07	1.43–3.00	2.04	1.41–2.96	2.04	1.41–2.96
Mortality						
Variable	Non-Adjusted Model		Model 1		Model 2	
VH-SWI	1.00	---	1.00	---	1.00	---
H-SWI	1.59	1.27–1.99	1.57	1.25–1.97	1.54	1.22–1.94
M-SWI	1.42	1.15–1.76	1.41	1.14–1.75	1.41	1.13–1.76
L-SWI	1.05	0.81–1.36	1.04	0.80–1.35	1.02	0.78–1.33
Recovery						
Variable	Non-Adjusted Model		Model 1		Model 2	
VH-SWI	1.00	---	1.00	---	1.00	---
H-SWI	1.00	0.77–1.30	1.03	0.79–1.35	1.06	0.81–1.40
M-SWI	0.63	0.49–0.80	0.60	0.43–0.77	0.59	0.46–0.76
L-SWI	0.74	0.56–1.00	0.73	0.54–0.98	0.73	0.54–1.00

Note: VH-SWI—Very High Social Welfare Index, H-SWI—High Social Welfare Index, M-SWI—Medium Social Welfare Index, L-SWI—Low Social Welfare Index. Model 1: Adjusted by age and sex, Model 2: Adjusted by age, sex, number of comorbidities and smoking.

## Data Availability

Reported data are available at https://datos.cdmx.gob.mx/dataset/base-covid-sinave/resource/ede8e4df-02cb-459f-ab29-78a0610c99c8 (accessed on 10 February 2021).

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
