# Peer review of "Epidemiological Profile and Social Welfare Index as Factors Associated with COVID-19 Hospitalization and Severity in Mexico City: A Retrospective Analysis"

_ijerph, 2022, doi:10.3390/ijerph192214803_

Round 1
Reviewer 1 Report
Many other studies have shown similar associations, so the authors need to clarify what is added by this study. If the setting is unique, expand on why this is a particular interesting or informative case. Authors mention significant administrative and socio-economic differences across the 16 regions. A deeper exploration of this would be a bigger contribution; how do associations vary by region, and what are some potential contextual explanations?
Avoid causal language: change wording like “may impact”, to “are associated with”.
Abstract reports Mexico as 20th place for cumulative cases, but introduction reports 18th place.
Figure 2 Panel B is difficult to read due to the large number of regions. Consider instead highlighting a few regions of interest or using different panels for each SWI category.
Since individual cases are the unit of analysis, why are there so few in the dataset? Needs a more in-depth discussion of sample selection and any limitations of the dataset used.
Reviewer 2 Report
Reviewer Comments
1-In section 2.2 “Data Collection”, the authors mentioned that the data of the present study were retrieved from the Virology Analysis and Reference Unit Database. It is very important to understand how the researcher obtained the sample. However, information about sampling method(s) was not introduced. So, manuscript authors should always provide sufficient methodological detail to demonstrate the utility of their desired and obtained sample sizes, carefully describing the impact of sample construction on generalizability of results. Also, there is no relevant information about whether the real data has a nested structure. For example, patients are nested within hospital, hospitals are nested within major regions within Mexico City. Please provide relevant information. Furthermore, I am curious if the real data has missing values. If it is the case, then what is the missing rate and pattern, respectively. If not, please explain to the reader in one sentence.
2-In line 89, the authors mentioned that the present study is a retrospective study. However, the necessary information about the variables used in the study are not provided. I strongly suggest authors write a short paragraph to clearly introduce the outcome variable, independent variables (covariates), and/or confounders.
3-Please combine the paragraph of 2.3 Data Analysis and the paragraph of 2.4 Statistical Analysis and you can use a subtitle like “Data Analytic Strategy”.
4-In line 129-130, the authors mentioned logistic regression model was used to determine the association between the SWI categories and COVID-19 clinical outcome. However, several important information about logistic regression model is missing. Please answer the following questions,
What estimation method did the researchers use in the present study to estimate the parameters of logistic regression model? (Maximum Likelihood or other methods?)
Overdispersion issue is not uncommon in logistic regression model for binary response variable. How did the researchers investigate this issue and identify potential factors that contribute to potential overdispersion (e.g., sampling design, and omitted variables).
In essence, logistic regression belongs to linear model, also known as log linear model. So, researchers should determine that the relationship between independent variables and outcome variable (logit p) is linear. If not, possible transformations should be done for smooth implementation of the model. Otherwise, it will be wrong. How did the researchers justify the appropriateness of this linearity? Have you ever tried the nonlinear term when fitting the logit models? Also, information about model fit and measures of association should be provided. Note, good model fit in logistic regression is best assessed through a collection of evidence rather than relying on a single criterion. (You can consider the following indices such as Hosmer-Lemeshow test statistic, McFadden’s R2, AIC, BIC, and Deviance etc.)
5-In line 174 and line 194, the authors used the word “probability”, which is incorrect. Because you fitted logistic regression model in the present study, and the results are presented in Table 3 in the metric of “OR” (odds ratio). Why do you say “probability” here? You can say something like “risk”. If you really want to convert odds ratio to probability, you can use “margin” command in Stata to do this.
6-The authors should pay much attention to English writing because sometimes I found it difficult to follow the author’s argument due to the many stylistic and grammatical errors. Due to the limited time and space here, I just list primary issues found in the text,
*Issues in the text
Line 2-4 The first letter of each word in the title of the article must be capitalized.
Line 73 “lack in unfollow…” ―know that lack in as a noun is substandard English.
Line 78 “in one hand” ―There is no such thing in English. Correct: on one hand
Line 81 “mayor regions” ― I cannot follow the sentence and guess it should be “major”.
Line 84-86 The logic of this paragraph is problematic. “Since …, however….” Should be changed as “although …., remove (however), whether…).
Line 90 I.S.S.S.T.E―Author should illustrate what does this abbreviation mean.
Line 112 ”divided in” ― It should be “divided into”.
Line 132 139 190 191 “p” ― They should be italicized. (p)
Line 164 “Figure 3” ― Where is Figure 3? I cannot find it at all.
Line 191 “Table 3” ― 23.7% and 25.8% are not there. Please double check your writing.
*Issues in the figures
Figures 1 and 2 are so blurry that the illustrative effect is affected. Also, the legend, subtitle, and title of these two figures need to be adjusted so that readers can understand the meaning that they convey.
*Issues in the tables
There is no need to highlight any column or row in the tables in this article.
The word “Note” is missing at the front of the text annotated at the bottom of each table.
Please remove the parentheses and annotated texts about SWI at the bottom of each table.
Please align all decimals in each table.
Reviewer 3 Report
The Manuscript „Epidemiological profile and social welfare index as factors associated with COVID-19 hospitalization and severity in Mexico City: a retrospective analysis“ deals with an important issue which was additionally underlined by the COVID-19 pandemic – the inequalities in health and how unjust they can be. The Authors have explored, in a sample of almost 4000 positive cases, the association between the social welfare index (SWI) and hospitalization, mechanical respiratory support, and death, and found that the SWI was associated with these outcomes – lower SWI associated with a higher probability of a negative outcome.
The topic is really important, but this Manuscript needs both scientific re-writing and re-formatting, as well as an English language revision. After that, I would recommend it for publication.
Abstract
The abstract should try to provide some concrete background information with actual numbers (number of cases per 100k, number of deaths per 100k, comparison between Mexico and the region or world).
The Result part of the abstract could be clearer. What analysis was done? ODDS RATIO? If yes, please show odds ratios and confidence intervals considering the outcomes and low vs medium vs high (as already done, just show the whole measures of effect).
It is not clear in the abstract if univariate or multiple logistic regression was done. Is SWI associated even when other variables are included in the model. Please clarify.
Try to add one phrase about the statistical method.
Introduction
Please exclude all the info which has become the general knowledge (the whole first paragraph, except for the last phrase – add a date when these numbers are accurate).
Lines 67-69: This phrase is unclear, please rewrite it.
A background research regarding the socio-economic factors and their influence on the COVID-19 outcomes (disease, severe disease, hospitalizaiton, respirator, death) is missing. The Authors need this, to present previous research and to show the knowledge gap they will close with their research.
Methods
It is unclear if SWI was calculated for each case, or it was taken based on location.
It is also unclear if the cases (diagnostically confirmed) are in the database, or all suspect cases. Maybe a chart explaining the data entering or exiting the analysis would be helpful.
Please rewrite: Figure 1. Epidemiological behavior of COVID-19 cases (it cannot be epidemiological behavior).
Was Ethics committee consulted?
Results
Please provide all numbers in a standardized form (e.g. XX per 100,000 inhabitants) so that numbers can be compared between the townhalls.
The part about the diagnosis should go into the methodology.
The results are well presented, especially the models. It would be useful to highlight in the tables what is Model 1 and Model 2 – I see it is written below but maybe there is a better way to show this info.
Why are not all comorbidities specifically included in model 2? Could a specific comorbidity or diagnosis be associated with the negative outcome more than SWI?
Discussion
I propose that the Authors follow the presentation from the STROBE guidelines (https://www.equator-network.org/reporting-guidelines/strobe/).
Concentrate on other studies exploring socio-economic association with COVID-19 outcomes – what is the same, what is different, what knowledge have you closed?
Conclusions
You have confirmed that SWI is associated.
Is there an intervention? What could be done to practically use this information?
Round 2
Reviewer 3 Report
I congratulate the Authors for improving their Manuscript and increasing its value for the readers of IJERPH.